# Interests of the Non-Human Primate Models for HIV Cure Research

**DOI:** 10.3390/vaccines9090958

**Published:** 2021-08-27

**Authors:** Gauthier Terrade, Nicolas Huot, Caroline Petitdemange, Marie Lazzerini, Aurelio Orta Resendiz, Beatrice Jacquelin, Michaela Müller-Trutwin

**Affiliations:** 1Institut Pasteur, Unité HIV, Inflammation et Persistance, 75015 Paris, France; gauthier.terrade@pasteur.fr (G.T.); nicolas.huot@pasteur.fr (N.H.); caroline.petitdemange@pasteur.fr (C.P.); marie.lazzerini@pasteur.fr (M.L.); aurelio.orta-resendiz@pasteur.fr (A.O.R.); michaela.muller-trutwin@pasteur.fr (M.M.-T.); 2Université de Paris, Bio Sorbonne Paris Cité, 75013 Paris, France

**Keywords:** animal model, non-human primate, HIV, SIV, natural host, cure, therapies, immunity

## Abstract

Non-human primate (NHP) models are important for vaccine development and also contribute to HIV cure research. Although none of the animal models are perfect, NHPs enable the exploration of important questions about tissue viral reservoirs and the development of intervention strategies. In this review, we describe recent advances in the use of these models for HIV cure research and highlight the progress that has been made as well as limitations using these models. The main NHP models used are (i) the macaque, in which simian immunodeficiency virus (SIVmac) infection displays similar replication profiles as to HIV in humans, and (ii) the macaque infected by a recombinant virus (SHIV) consisting of SIVmac expressing the HIV envelope gene serving for studies analyzing the impact of anti-HIV Env broadly neutralizing antibodies. Lessons for HIV cure that can be learned from studying the natural host of SIV are also presented here. An overview of the most promising and less well explored HIV cure strategies tested in NHP models will be given.

## 1. Introduction

Forty years after the first clinical observations of Acute Immunodeficiency Syndrome (AIDS) and thirty-eight years after the identification of HIV as the etiologic cause of AIDS [1], a vaccine and a cure are still lacking. Relentless research has been conducted, leading to major scientific and therapeutic breakthroughs, one of the most prominent being the introduction of antiretroviral therapy (ART). ART can permanently suppress HIV replication. However, even today, only 73% of people living with HIV (PLWH) have access to ART [2]. Moreover, ART is not curative, must be taken for life and does not target HIV-associated chronic immune activation.

A safe, effective and durable way of completely eliminating HIV infection (eradication referred to as “HIV cure”) or suppressing viremia in the absence of antiretroviral therapy (“HIV remission”), would likely play a critical role in controlling AIDS epidemic and ending it [3]. Hence, research and longstanding efforts to develop curative approaches need to be pursued. In this review, we will use the word “cure” irrespective of whether the aim is to achieve eradication or remission.

Animal models have been the lynchpin of a tremendous amount of therapeutic and preventive advances in modern medicine. In the field of HIV science, they played a central role and helped to understand the infection and develop the therapies currently available [4,5]. The non-human primate (NHP) models for HIV/AIDS include several macaque species infected with various Simian Immunodeficiency Viruses (SIV), as well as natural hosts of SIV. Of note, NHP models have some inherent limitations at various levels and their use must be done with caution and appropriate monitoring. Nevertheless, they are so far the only animal models to fully reproduce the pathogenesis induced by HIV in humans, and therefore are valuable tools for the development of therapeutic strategies towards an HIV cure despite their limitations [6]. 

In this review we aim to present why animal models are still needed in HIV science and the contribution of NHP models in HIV cure research. We will highlight some examples of results relevant for HIV cure strategies obtained using NHP and how their use in HIV science contributes to the search of new innovative therapies. We will focus on the contribution and limitations of NHP models, other valuable animal models such as humanized mice being reviewed elsewhere [7,8,9,10].

## 2. Could Studies in Humans and In Vitro Models Be Sufficient for HIV Cure Research?

The study of new therapeutic strategies to cure HIV have involved, inter alia, the longstanding contributions of animal models. These models enable pharmacological and preclinical trials to assess the pertinence of a curative strategy in vivo. However, the results obtained in NHP were often not reproduced in humans or gave mixed results [3]. In addition, with the technological improvements, some might consider that the use of NHP models is no longer necessary in HIV research and in particular not ethically justified. Indeed, other valuable tools are used in HIV science, such as model cell lines, primary cells from PLWH, and even tissue explants and organoids, which allow some pathophysiological mechanisms to more closely reflect in vivo conditions. Nevertheless, the in vivo complexity of an entire body can only be achieved by using animal models. Animal models should be used in the most parsimonious way in line with the 3Rs principles (Replacement, Reduction and Refinement). Yet, the major obstacles in HIV cure research are unlikely to be overcome without additional contributions from animal research. These challenges include: (i) The wide diversity of HIV infected cells, not all of which are well-defined and are dependent of complex anatomical and immunological (micro) environments. (ii) The difficulty in selectively targeting and distinguishing latently infected from uninfected cells. (iii) The occurrence of HIV persistence in multiple tissues, sometimes hardly accessible (deep reservoirs). (iv) The impairment of immune responses during the course of HIV Infection and HIV-associated chronic immune activation and tissue damage.

The use of animal models to better understand and overcome these barriers provides some advantages that are not reflected in clinical trials and in vitro models. They allow to control the time, dose, inoculation route, host genetic factors, and virulence and genetic diversity of the infecting virus. This provides a high degree of experimental control and thus fewer unpredictable/unknown confounding variables than is typically achievable in human clinical studies. They provide a critical tool to understand the complex virus–host interactions and immune responses, that cannot be accurately modeled in cell cultures or other more reductive systems [6,11]. Another key advantage of studies performed on animal models is the possibility to perform longitudinal analyzes with access to pre-infection baseline, as the different parameters studied can be measured prior to inoculation [6].

Also, animal models enable to study the very first hours of infection, including the ability to initiate ART very early after infection. Moreover, with appropriate animal welfare monitoring and veterinary support and expertise, they allow to routine and longitudinal collection of fluids and biopsies with greater frequency than is possible in human studies (including blood, cerebrospinal fluid, bronchoalveolar lavages, and bone marrow aspirates). They also provide a means to sample tissues that are difficult or impossible to collect in humans for logistical and ethical reasons, but could represent key sites for understanding viral replication and pathogenesis (e.g., vaginal and rectal mucosa, LNs, spleen, liver, lung and brain) (Figure 1) [5,6,12].

In addition, animal models allow the monitoring of the viral load for several weeks or months after analytical treatment interruption. Not restarting ART in case the viremia is too high would often be considered too risky for viral transmission in humans and effective risk mitigation would be required that limit the interpretation of the study [13].

However, it should be stressed that many results obtained in animals cannot be directly translated to humans. Indeed, the results obtained in NHP were not necessarily confirmed in further human studies for several reasons. First, one must keep in mind that, despite numerous similarities, there are always significant genetic and physiological differences between a given animal species and humans. These differences must be thoroughly described and understood to be considered in experimental design and interpretation of observations. While challenging, they may ultimately lead to the discovery of new mechanisms and allow us to envision innovative therapies. A second reason that may explain the difficulty of extrapolating results obtained in animals to humans is the fact that the animals are most often not treated for long time with ART. The reservoir size and composition strongly vary depending on the time of initiation and duration of ART. In many cases, the ART schedule between NHP and PLWH were not the same which could explain some of the differences observed. Also, the homogeneous genetic composition of the animals used in many laboratory models, which in those cases do not reflect the genetic and physiological variations within each species. There is no animal model capable of fully mimicking a given human disease, which itself is polymorphic between patients. Yet the differences that characterize animal species and strains can represent remarkable opportunities to understand different host responses, disease development and various host-pathogens interactions (especially in the case of infectious diseases), which may actually benefit the research for new cures [14]. It is worth noting that animal research has contributed to undeniable medical advances. Of the 111 Nobel Prizes awarded for Physiology or Medicine, 98 were dependent on research involving animals, including every single prize awarded for the past 30 years [15]. In addition, preclinical testing of new drugs still relies heavily on animal testing, with only an estimated 10% of new drugs successfully pass to acceptance for clinical use. It seems impossible in this context to currently replace animal testing in the absence of alternatives [14,16].

The implementation of tools to better understand the genome and immune responses of NHPs, as well as the advent of new recombinant viruses, have led to increased relevance of available animal models. Although none of these models exactly match HIV infection in humans, each model can be valuable to address the key specific questions in HIV cure research. It always remains essential to define the benefits and limits of each proposed project, as well as continuing the use of the 3Rs rules while conducting animal research [4,5,17]. Other valuable tools for basic, translational and preclinical research, such as tissue explants and organoids, need to be further developed in parallel to be used in a complementary manner and to replace the use of animals, when possible, even though the use of some animal models will likely still be necessary to link to clinical research and address specific issues. Recent advances in technology, such as new imaging techniques, are also likely to improve research while allowing a reduction in the number of animals required [4,5,17,18,19]. Overall, the use of NHP is already very low (<1%) in comparison to other animal models. As shown by the Covid-19 pandemic, the demand for NHP to develop vaccines and drugs against a new human disease might temporarily increase.

At last, it should be mentioned that animals can also benefit from animal research, whether directly through veterinary medicine, with 90% of the veterinary drugs used to treat animals being identical or highly similar to those used to treat humans [14], or through a better comprehension of diseases that affect animals, which can be used for conservation purposes in the wild. For example, it has been shown that chimpanzees (*Pan troglodytes troglodytes*) infected with SIVcpz, the ancestor of HIV-1, also suffer from decreased lifespan and AIDS-similar immunopathologies. This discovery, which can contribute to understand the population dynamics stakes of this endangered species, has been permitted by noninvasive samplings and observations in the wild but also benefited from the general knowledge on lentiviruses that was gained using animal models [20,21,22,23]. These important bioethics considerations are widely discussed elsewhere [16,24,25,26].

## 3. Animal Models Used for Research on HIV Cure

### 3.1. General Considerations

The only animal species susceptible to HIV viruses are NHPs. Among NHPs, Asian monkeys (macaques) infected with viruses close to HIV-2 (SIVmac) are the most commonly used. Inoculation of SIVmac in the latter leads to persistent infection and progression to AIDS, generally in a similar manner to PLWH [4,27]. Almost all the pathophysiological features of HIV-1 pathogenesis in humans are recapitulated in this model, including viral targeting of CD4+CCR5+ T cells, very early depletion of CD4+ T cells at mucosal sites, massive CD4+ T and Th17 cell depletion in the gut in the very early phase of infection, and associated damage to the intestinal epithelium, translocation of microbial products from the gut lumen into the blood stream, chronically elevated levels of immune activation, progressive depletion of CD4+ T cells in the periphery, and eventual development of opportunistic infections and malignancies [4,6].

The implementation of an animal model for the search of an HIV-1 vaccine still presents a significant difficulty. This difficulty is due to the fact that HIV-1 tropism is strictly limited to humans and chimpanzees. The first animal models developed and used in the fight against HIV therefore involved chimpanzees, a model that is now abandoned for ethical reasons [18,28].

Historically, feline models using cats infected with immunodeficiency virus (FIV) also served as models of naturally occurring immunodeficiency [29]. However, because FIV infection does not reproduce all the features of HIV infection and some important aspects differ markedly between cats and humans, this model is rarely used [4,30]. Mice, rats, or rabbits were originally proposed, but they could not be successfully infected with HIV [4,31,32]. In order to circumvent this, humanized mouse models that can be infected by HIV have been developed [7,8,9,10]. These models can be used to answer specific questions, such as testing the inhibitory potency of HIV replication by specific molecules. However, they do not reflect the complex pathophysiological virus-host interactions and the characteristics of long-lasting persistent viral reservoirs in contrast to NHP models.

### 3.2. The Use of Macaques as a Model of HIV Infection

Macaques, once infected with SIVmac, can reproduce all the distinct outcomes of infection observed in humans: Some animals can progress rapidly (in a few months) towards an AIDS-like disease, while the majority progress to disease after a period of several months/years of chronic infection, and finally, a minority of macaques may spontaneously control HIV infection, similar to what is observed in human “elite controllers” or ”HIV controllers” [17,33,34]. Such viral control is, as in human controllers, often associated with specific MHC alleles in the controller macaques.

Two macaque species are mainly used as models in HIV science: rhesus macaques (RM, *Macaca mulatta*) and cynomolgus macaques (CM, crab-eating macaque, *Macaca fascicularis*). The pigtailed macaque (*Macaca nemestrina*) is used less frequently. Each species displays a distinct susceptibility to SIVs and disease progression profile, which also varies according to the geographical origin of the animal. The most frequently used RM are of Indian and Chinese origin [5,17,35]. CMs display variable levels of viremia and disease progression, depending on the host, the viral load and route of infection. When infected intrarectally with a high viral load of SIVmac, the viremia levels and disease progression profiles are similar to that observed in Chinese RMs or humans [36]. However, spontaneous control is observed more frequently in CMs than in RMs and humans. CMs used in experimental research often originate from the island of Mauritius. Mauritius has a population of CM descended from a small number of founder animals that have been geographically isolated from other CMs since their establishment. As a result, this population of animals has limited MHC class I and II diversity and has become an intriguing tool for studies seeking to better explore MHC haplotypes than would be possible in other, more outbred NHP populations [6,17,37,38,39,40]. Pigtailed macaque, when infected with SIVmac, can represent a model for rapid progression to AIDS, with disease progression often occurring within three months [41,42]. However, in macaques, and in monkeys in general, the diversity of MHC sub-combinations, as well as other immune components, such as the various Fc receptors, KIR genotypes and interferon-alpha subtypes remain poorly characterized for the time being and require further investigation.

Thus, the chosen model may vary according to the species and origin of the macaque, as well as the virus strain used and the experimentally controlled parameters of infection (viral load, route, time, …). In this respect, what can be derived from a macaque model is highly dependent on the parameters used, and these must be wisely set up according to the question addressed.

### 3.3. Viruses Used to Infect NHP Models of HIV Infection

SIVs have been identified in more than 40 African NHP species. Only African NHPs carry SIV in the wild. NHPs and SIV might have co-evolved up to one million of years [4,17,43]. The first species identified as a natural SIV carrier was the sooty mangabey (SM, *Cercocebus atys*), which can carry the virus SIVsmm [44]. African green monkeys (AGM, *Chlorocebus sabaeus*) and mandrils (*Mandrillus sphinx*) can also be infected with SIVs in the wild (SIVagm, and SIVmnd-1 & -2, respectively) [4,17]. These NHPs represent a large reservoir for lentiviruses [45,46], and display a very high seroprevalence (up to 45–50%) [43,47]. 

Because RMs of Indian origin were the first recognized non-natural hosts of pathogenic SIV infection, the initial development of SIV for animal models involved serially passaging virus from animal to animal within RMs of Indian origin. The SIVs thus obtained were called SIVmac [6] but came originally from the transmission of an SIVsmm to RM. Most of Indian-origin RMs develop high peak viral loads (generally over 10^7^ viral RNA copies per mL of plasma) and sustained high chronic viral loads when infected with either the uncloned viral isolate SIVmac251 or the infectious molecular clone SIVmac239 [48]. A more rapid disease progression is generally observed with SIVmac239 compared to SIVmac251. Some other (rarely used) SIVmac strains display lower virulence associated with attenuated infections in RMs [4,49,50]. 

SIVsmm and SIVagm viruses are non-pathogenic in their respective natural hosts, but RMs and pigtailed macaques develop AIDS-like disease after infection with certain SIVsmm or SIVagm strains. SIVagm.sab92018 infection in RMs results in something similar to what is observed with HIV infected “elite-controllers” in humans; and consequently, SIVagm.sab92018-infected RMs have been proposed as an animal model for Elite-Control of HIV infection [51]. Several SIVs can thus be used to infect animal models, with distinct degrees of virulence. The outcome of SIV infection in NHP is then determined by the combination and complex interactions between viral and host determinants (Figure 2) [4,5,51,52,53,54,55,56].

Vaccine studies are hampered by the fact that macaques are not susceptible to HIV-1 infection. In particular, studying the efficacy of neutralizing antibodies depends on the ability to challenge the animals with viruses encoding the HIV-1 ENV gene. To overcome these limitations, chimeric simian-human immunodeficiency viruses (SHIVs) were developed in which SIVmac ENV was replaced with HIV-1 ENV. Similarly, to overcome resistance of SIVmac to some drugs (NNRTIs) for use in RMs and pigtailed macaques, SHIVs encoding for the HIV-1 reverse transcriptase (RT) gene were engineered, such as RT-SHIVmac239 and RT-SHIVmne. RT-SHIVs still have their limitations, including initial difficulties in suppressing their replication with the same ART treatments used in humans (i.e., tenofovir/emtricitabine/ efavirenz) [5]. In addition, SHIV infections in general have limitations in that they do not fully reproduce the physiopathology of HIV-1 and SIVmac infection. Thus, the infection is more often attenuated and the dynamics of CD4+ T cell subsets are different.

### 3.4. Essential Insights on HIV Infection Obtained with the Help of NHP Models

Some of the past and current advances in the field of HIV could not have been obtained without animal models [57]. Here we list some of the major contributions of animal models to HIV/AIDS research.

For instance, important proofs of concept for preventive or curative therapeutic approaches have been provided by studies in animal models, such as the ability of neutralizing antibodies to protect against infection [58]; or the ability of broadly neutralizing antibodies (bNAbs) and of cytomegalovirus-vectored vaccines to induce a control [59,60,61,62,63,64]; or other curative strategies developed in Part 5. 

The cellular targets of HIV and their dynamics in blood and tissues have also been better understood thanks to NHP models, including: The fact that resting memory T cells are a major target of the virus in lymphoid tissues [65,66], the fact that loss of central memory T cells is associated with disease progression [65,67], a better understanding of the trafficking of Treg, plasmacytoid dendritic cells, and NK cells to the gut [68,69,70] and the rapid and dramatic depletion of CD4+ T and Th17 cells in the gut [71,72]. Humanized mouse models also proved helpful to study the ability of myeloid cells to support latent HIV infection and supported the possibility of a myeloid reservoir for HIV [73,74,75].

NHP models contributed to the understanding that adaptive immune responses arrive too late at the site of early viral replication. Studies in macaques have shown that upon infection at mucosal sites, the virus crosses the epithelial barrier by several distinct mechanisms, rapidly establishes initial founder populations of infected cells (foci) and subsequently spreads into draining LNs [76]. The concepts that CD8+ T cell responses arrive “too little” and “too late” in the mucosa to control viral dissemination and that the “window of opportunity” to prevent infection is very short were developed in studies performed on macaque models [77,78,79].

The impact of CD8+ T cells on viral set-point and the association of HLA/MHC-I genotypes with rapid or slow AIDS progression has also been elucidated with the help of animal studies, in particular by studying CD8+ T cell responses during the first days after infection, which was not possible for a long time in humans [80]. 

## 4. The Case of Natural SIV Hosts and Their Potential Use for HIV Cure Research

AGM, SM and mandrill are the three SIV natural hosts species commonly used in HIV research [81] and represent models of non-pathogenic infection. The natural habitat of SMs corresponds to West-Africa and that of mandrills to Central Africa. AGMs are present throughout sub-Saharan Africa except tropical forests and constitute the largest reservoir of SIVs [43]. SIVmnd-infected mandrills share a number of features with SIV-infected AGMs and SMs and all three generally do not progress to disease [82]. Natural hosts of SIV share some common features that delineate their specific infection pattern:(i)First, natural hosts exhibit high levels of viremia, similar to those observed in untreated pathogenic SIV infections of RMs and HIV infections of humans both in primary and chronic infection [82,83,84,85,86,87]. During chronic infection, the virus thus continues to replicate at high levels, in most cases to about 10^4–^10^7^ SIV RNA copies/mL of plasma, without immune damage [85]. These characteristics are similar to the “viremic non-progressors”, very rare human individuals who show high viremia but maintain CD4 T cell counts and avoid disease progression for years [88].(ii)Natural hosts avoid chronic immune activation, which is the driving force of CD4+ T cell depletion and progression to AIDS in humans [84,89,90,91,92,93]. SIVagm and SIVsmm infections trigger a potent type I-interferon (IFN) production in acute infection, but this inflammatory response is rapidly controlled [94]. After the acute phase of infection, immune activation is controlled and returns to near pre-infection levels [89,94]. This could be of relevance for the lack of intestinal tissue damage (see below).(iii)Natural hosts show strong viral control in secondary lymphoid tissue (SLT) [95,96]. Indeed, AGM and SM exhibit a strong control of viral replication in lymph nodes (LN) (in both T cell zone and B cell follicles) shortly after the peak of viremia, which persists throughout infection, despite high viremia levels. Viral control in SLT of SIVagm-infected AGM is mediated by NK cells [97]. The latter express C-X-C chemokine receptor type 5 (CXCR5) in SLT during SIVagm infection and are able to migrate into B cell follicles [95,96,97]. This represents a striking difference with pathogenic infections, where the virus persists in LN and where B cell follicles represent “sanctuary sites” for the virus. Immune activation, including expression of IFN-stimulated gene (ISGs), is particularly rapidly controlled in SLT. The rapid and strong viral control most likely contributes to this rapid resolution of inflammation. The LNs of natural hosts also show neither lymphadenopathy nor fibrosis. Importantly, the network of follicular dendritic cells (FDC) remains intact, unlike in HIV-1 and SIVmac infections [95]. During SIVagm infection, these FDC produce high levels of IL-15. NK cells accumulate preferentially in these IL-15+ follicles during SIVagm infection [97].(iv)Central memory CD4+ T-cells (T_CM_) in SMs have been reported to be infected at a lower frequency than in non-natural hosts. Based on this observation, it has been suggested that long-lived T_CM_ cells are relatively resistant to SIV infection. Indeed, it has been shown that in sooty mangabeys, T_CM_ exhibit low levels of SIV co-receptor CCR5 expression and are less likely to be infected in vivo and in vitro (compared with sooty mangabeys effector memory CD4+ T cells and RM central memory CD4+ T cells) [98]. However, SIVsmm and SIVagm do not require CCR5 to infect CD4+ T cells but can efficiently use other co-receptors, such as CXCR6. Thus, the underlying mechanism of the lower infection rate of Tcm could be another one. For instance, the fact that SIVsmm infection is strongly controlled in SLT but not in the intestine might play a role since the frequency of T_CM_ is higher in SLT than in the intestine. Whatever the mechanism, the preservation of long-lived cells in lymphoid tissues in natural hosts can contribute to the reduced pathogenicity [83,98,99,100].(v)Natural hosts of SIV preserve their intestinal mucosal immune system and the integrity of the intestinal barrier. Thus, they efficiently prevent the translocation of microbial products from the intestinal lumen into the systemic circulation [101,102]. In addition, no early preferential depletion of Th17 cells is observed during SIV infection of natural hosts [71,103,104]. Th17 maintenance in the gut could positively contribute to preserve the intestinal barrier integrity [71]. A recent study [105] by Raehtz et al. documenting early SIV infection of AGMs showed that despite a strong but transient interferon-based inflammatory response, the levels of plasma markers associated with enteropathy did not increase. They did not document a significant increase in apoptosis of mucosal enterocytes or lymphocytes, nor damage to the mucosal epithelium [105]. Also, unconventional CD8+ T cells expressing regulatory molecules expand in the intestine of SIVagm-infected AGM and the increase of these cells was associated with lower levels of intestinal inflammation as measured by IL-23 [106]. It is also possible that stronger or more efficient tissue repair mechanisms operate in natural hosts of SIV. Barrenas et al. [107] demonstrated that monocytes from AGMs rapidly activate and maintain evolutionarily conserved regenerative wound healing mechanisms in mucosal tissues, possibly via fibronectin production and TGF-beta signature.

Because of the high viremia levels characterizing SIV infection in natural host, it is often assumed that natural hosts could not represent models that contribute to the search for an HIV cure. Indeed, a similar absence of disease progression but with such high viremia levels cannot be tolerated in humans because of the high risk of viral transmission. Nonetheless, we propose that natural hosts may represent excellent models to unravel the strong viral control in SLTs (in contrast to PLWH in which SLTs constitute important HIV reservoirs) and the mechanisms of protection against tissue damage, microbial translocation, and undue immune activation. A better understanding of these mechanisms of protection are relevant for HIV cure research, because tissue damage and systemic inflammation contribute to the exhausted and weakened anti-HIV innate and adaptive immune responses, in particular in those individuals who initiated ART in chronic infection, which still are the majority of the PLWH. For instance, uncovering the mechanisms of Th17 cell preservation during natural SIV infection could enable the identification of new therapeutic targets to improve Th17 cell homeostasis in PLWH, thereby promoting immunological restoration of the intestinal mucosal barrier. A better understanding of the mechanisms involved in the control of SIV-infection in LN and spleen of natural hosts could guide the development of novel strategies to control HIV-associated in these sanctuaries. It could also prove very useful for a better understanding of the regulation and impact of early anti-viral innate immune responses. Thus, investigating the mutually beneficial co-adaptations between primate lentiviruses and their natural hosts that have co-evolved over hundred thousands of years, might prove to be insightful also for HIV cure research [17,43,83].

## 5. Evaluation of HIV Cure Strategies with the Help of NHP Models

NHPs not only serve as models for the development of vaccine candidates but have also been used as preclinical models to develop and evaluate many of the strategies designed to achieve an HIV cure.

### 5.1. Latency Reversing Agents for “Shock and Kill Strategies”

One of the main strategies tested for achieving a HIV cure is often referred to as “shock and kill” and involves the use of latency reversing agents (LRAs) to induce viral gene expression in infected cells. These cells expressing viral RNA and proteins would either be exposed to viral cytopathic effects or become recognizable targets for the immune system or host-specific therapies [108]. NHP models have been used to assess the potential and limitations of various drugs as LRAs.

#### 5.1.1. Epigenetic and Signal Agonist LRAs

A classical approach to reverse HIV latency is to target epigenetic modulatory mechanisms. NHP models can help to identify epigenetic modulators that can be used as novel LRAs. For instance, histone crotonylation was recently shown to be a regulator of HIV latency that could be targeted for therapeutic purposes. Indeed, in an NHP model of HIV/AIDS (SIV-infected RMs), reactivation of latent HIV was observed after an increase in expression of the fatty acid metabolic enzyme ACSS2 (crotonyl-CoA–producing enzyme acyl-CoA synthetase short-chain family member 2), which induced histone lysine crotonylation [109]. Several histone-deacetylase inhibitors (HDACs) have been tested, but overall have not shown promising results.

Another example of LRA is provided by the increasingly studied activators of the non-canonical (nc) NF-κB pathway. The ncNF-κB pathway is activated by the second mitochondrial-derived activator of caspases (SMAC), which inhibits the inhibitor of apoptosis proteins (IAPs) [110,111]. It has been shown in vitro that SMAC mimetics, such as AZD5582, can reverse this suppression in CD4+ T cells, potentially reactivating the latent reservoir [112]. Using an ART-suppressed HIV-infected humanized BLT mouse model, Nixon et al. showed that AZD5582 treatment can induce systemic HIV RNA production in vivo. They complemented their findings in the mouse model by using an NHP model in which AZD5582 induced SIV RNA expression in the plasma and LNs of ART-suppressed SIV-infected RMs. Their experiments demonstrated that HIV or SIV reactivation could be robustly induced by SMAC mimetics, although no consistent reduction in the size of replication-competent reservoirs was observed [111]. In a study by Dashti et al. which also used an NHP model to assess the efficacy of the same SMAC mimetic AZD5582 in combination with an antibody-derived DART molecule, AZD5582 apparently did not induce sufficient latency reversal [113] while Mavigner et al. recently showed that the effect of AZD5582 could be enhanced by CD8α cell depletion [114]. The ability of SMAC mimetics to reproducibly reverse latency in vivo needs to be better characterized and further investigated.

#### 5.1.2. Immunomodulatory LRAs

The ability of several Toll-like receptors (TLRs) agonists to reverse HIV latency has also been explored thanks to the use of animal models. Apart from this potential use, they may also offer other potential benefits for immunotherapies, as TLRs such as TLR7 and TLR9 can promote and enhance antiviral responses, promote antigen presentation, activate natural killer cells and memory B cells, and enhance anti-viral immune responses [115,116,117]. Agonists of TLR7 have led to interesting results in NHP models. In 2016, Borducchi et al. observed that the TLR7 agonist GS-986, in combination with therapeutic vaccination could lead to improved virological control, increased SIV-specific immune responses, and delayed viral rebound after ART discontinuation in SIV-infected RMs that began ART during acute infection [118]. In 2018, they also observed a delayed viral rebound in SHIV-infected macaques after administration of the TLR7 agonist GS-9620 (Vesatolimod) in particular when combined with an Env-specific broadly neutralizing antibody (bNAb PGT121) [119,120]. In the group of macaques receiving both GS-9620 and the bNAb PGT121, 5 out of 11 animals did not even experience viral rebound for more than 6 months after ART interruption. In these two studies, it is possible that the interesting effects of the two TLR7 agonists GS-986 and GS-9620 were not—or not only—due to a latency reversing effect, as transient increases in plasma virus were not clearly observed. Thus, other beneficial properties, such as immune enhancement and NK cell activation may have also come into play [119,121]. Indeed, in a computational model, the delay in viral rebound was best associated with NK cell activation. Another study by Lim et al. [122] investigated the ability of GS-986 and GS-9620 to induce transient viremia in SIV-infected RMs. Plasma viremia was successfully induced, and the results were consistent with a reduction in viral reservoir, in addition to the activation of multiple innate and adaptive immune cell populations. Of note, 20 to 25% of NK cells were activated after treatment [122]. Phase 1 clinical trials (NCT02858401 and NCT03060447) are already conducted to evaluate the safety of GS-9620 alone and in combination with other agents.

It has been shown in the macaque model, that persistently infected CD4+ T cells in LNs often express the Cytotoxic T-lymphocyte-associated protein 4 (CTLA-4) or Programmed cell death protein 1 (PD-1) [123]. These immune checkpoint blockers (ICBs) are generally employed to enhance immune responses. Recent studies in the macaque model have shown that anti-CTLA-4 combined with anti-PD1 act as potent LRAs [124]. Immune checkpoints blockade have therefore been proposed as a novel mean to reverse latency [125]. Multiple checkpoint blockade might be necessary to target the largest possible subsets of memory CD4+ T cells harboring replication competent provirus, including both CTLA-4+ and PD1+ CD4+ T cells [123,124]. Indeed, RMs receiving combined CTLA-4/PD-1 blockade exhibited greater numbers of reactivated viral lineages compared with anti-PD-1 monotherapy. The use of anti-CTLA4 and anti-PD1 or other ICBs in combination with killing strategies will be interesting to investigate in the future.

### 5.2. Block and Lock Strategies

Alternative strategies aim to deepen the latency state of the provirus so that the virus cannot be reactivated. This strategy is also called “block and lock” and some of them target the regulator protein Tat [126]. Studies are ongoing to decipher in more detail the factors impacting HIV latency, since these might be used later on for block and lock strategies. Molecules under investigation are for example LEDGIN [127] and those associated with HIV transcription patterns related to the circadian rhythm [128,129].

### 5.3. Immunotherapies to Elicit and Strengthen Potent Immune Responses

A variety of immunotherapies aimed at improving immune-mediated killing of infected cells have been tested in NHP models, including antibody-based therapies and therapeutic vaccines. 

#### 5.3.1. Broadly Neutralizing Antibodies and Beyond

Monoclonal antibodies nowadays play a major role in immunotherapeutic studies against many distinct types of diseases. Antibodies have long been the subject of HIV vaccine research and in the last decade became of major interest in HIV cure research. Both neutralizing and non-neutralizing antibodies have been analyzed in the past for their efficacies to eliminate infected cells expressing viral proteins. The discovery of the bNAbs has revolutionized the field [130,131]. It has been shown that the administration of a bNab potently reduces viremia. However resistant HIV strains arise relatively rapidly and thus the use of combination of bNAbs is preferred. Several complementary studies examined the effect of two bNAbs (3BNC117 and 10–1074). For instance, using an NHP model with SHIV-infected macaques, Nishimura et al. showed that early administration of these two bNAbs can reduce the levels of persistent viremia, establish T-cell immunity and result in long-term infection control [132]. The combination of these two bNAbs was subsequently evaluated in a phase 1b clinical trial (NCT02825797) [133]. To prolong the effect of bNAbs on viremia, novel generation of long-lasting bNAbs have been developed and are currently being evaluated. The use of vectors such as AAV expressing bNAbs also gave promising results (reviewed in [134]). The bNAbs seem to act through multiple mechanisms, including neutralization, antibody-dependent cellular cytotoxicity and CD8+ T cell vaccinal effect. However, alone they impact only little the viral reservoir and are thus currently investigated together with other molecules, such as TLRs or IFN-α2b. 

Antibodies targeting cellular epitopes of target cells could also be used therapeutically. Monoclonal antibody-mediated blockade of host-expressed α4β7 integrin has raised some interests in its ability to enhance virological control in NHP models [135,136]. Indeed, anti-α4β7 in NHPs could lead to lymphocytes redistribution in the body [137], alter the activation potential of α4β7-expressing cells, and interfere with virus binding to this integrin on target CD4 T cells [138], which together may account for the antiviral effects. However, the data obtained so far regarding their impact on post-treatment control are inconsistent and have not been confirmed in NHP models [136]. 

A biomarker that clearly defines cells latently infected by HIV or SIV is still missing. There might not be such single marker. However, several markers have been described that are more often expressed by HIV/SIV infected cells. Thus, HIV-infected CD4+ T cells often display a metabolic profile characterized by high glycolytic activity [139]. It has also been shown that CD32 expression is increased on CD4+ T cells that actively express viral RNA and that CD32+ CD4+ T cells increase in tissues after SIV infection [140]. This could be exploited to target and kill these cells. 

#### 5.3.2. Therapeutic Vaccines

Therapeutic vaccine approaches consist of inducing potent HIV-1-specific responses that could be harnessed for cure strategies. Traditional therapeutic vaccine approaches generally aim to potentiate adaptive immune responses and/or redirect HIV-1-specific CD8+ T cells. In the 2000s, using NHP models, it was shown that T cell vaccines delivering SIV proteins can induce cellular immune responses, reduce plasma viral loads, and preserve memory CD4 T cell counts in vivo [141,142,143]. However, except for the RV144 trial, none of the vaccine candidates tested in phase III trials in humans has shown a protective effect. In a recent study by Nakamura et al., SIV-infected ART-treated RMs were inoculated with Sendai virus vectors expressing SIV Gag and Vif. Gag/Vif-specific CD8+ T-cell responses were induced and became predominant. Even if viral rebound eventually occurred after ART discontinuation, these results suggest that induction of Gag-specific CD8+ T- cells by therapeutic vaccination may enhance anti-viral efficacy of CD8+ T cells. Although such a therapeutic vaccination strategy alone would not likely result in a functional cure, it could contribute to a more durable viral control under ART or in combination with other strategies [144].

In another vaccine study in the NHP model, Hansen et al. studied the effects of an SIV protein-expressing rhesus cytomegalovirus vector (RhCMV/SIV) [59,60,62]. They observed that immune responses elicited by the RhCMV vector were able to control SIVmac239 infection, whereupon the viral reservoir gradually decreased and the infection was eventually eliminated in some of these animals [60]. Subsequently, they showed that this replication-deficient CMV vector with SIV inserts could elicit and amplify an unconventional MHC-E-restricted TcR dependent CD8+ T cell response of unprecedented breadth, which (together with other factors) contributes to efficient control of SIV infection in half of the immunized macaques [62]. Furthermore, to limit the potential pathogenic effects of the RhCMV vector, the authors obtained its stable attenuation by deleting the RhCMV protein pp71 [63]. They investigated this vector in SIV-infected macaques and obtained similar efficacy to their previous studies. A clinical trial in humans is in preparation using an attenuated human cytomegalovirus (HCMV) vector following a similar approach [63].

As mentioned in the paragraph on the use of TLR7 as an LRA, Borducchi et al. observed improved virological control, increased SIV-specific immune responses, and delayed viral rebound after ART discontinuation in SIV-infected RMs that received a combination of TLR7 agonist and Ad26/MVA therapeutic vaccination [118]. Therapeutic vaccination was performed with a recombinant adenovirus serotype 26 (Ad26) prime and a modified vaccinia Ankara (MVA) boost. A therapeutic vaccine using Ad26 and MVA was recently studied in a clinical trial (NCT02919306). A heterologous vaccine regimen of trivalent Ad26 and MVA expressing two multivalent mosaic immunogens from the HIV proteins Gag, Pol and Env was administered to PLWH and generated robust immune responses. However, after ART interruption, the vaccine delayed the time to viral rebound by only a few days compared to placebo recipients, which was not statistically significant [13]. Nevertheless, these results are encouraging for therapeutic vaccines and combination strategies, and as pointed out by Mothe and Brander [145], we can wonder whether the lack of viral control might be related to still-insufficient stimulation of immune responses, inadequate response profiles, lack of reservoir mobilization, limited coverage of autologous viruses or expansion of T cell and B cell responses to irrelevant targets in the virus [145].

#### 5.3.3. Immune Checkpoint Blockers

Targeting ICBs and blocking them is increasingly being considered for therapeutic purposes in cancer and other diseases. There is the hope that ICBs might partially reverse HIV-associated immune dysfunction, enhance immune effector functions and HIV-specific CD4+ and CD8+ T cell responses in blood and tissues, and eventually lead to depletion of infected cells through Fc receptor activation or enhanced T cell killing, especially when used concomitantly with antibodies that activate Fc receptors [121,125]. However, they can also bear the risk of serious side effects.

Mylvaganam et al. studied the effects of an anti-PD-1 antibody in SIV-infected and treated macaques and observed decreased levels of cell-associated replication-competent virus, as well as beneficial effects on Th17 cell reconstitution in rectal mucosa, expansion of proliferating CXCR5+ effector CD8+ T cells, enhanced T cell responses, better control of viremia and reduction of the viral reservoir size [146]. This is consistent with a previous study by Velu et al. which suggested that administration of an anti PD-1 antibody to SIV-infected macaques can induce proliferation of memory B cells, increase in SIV envelope-specific antibodies and lead to significant reduction in plasma viral load [147]. 

Harper et al. demonstrated in SIV-infected long-term ART-treated RMs, that dual blockade of CTLA-4 and PD-1 can induce robust latency reversal, enhance T cell proliferation, and reduce the total amount of integrated virus. However, immune checkpoint blockade was not sufficient to achieve strong reservoir reduction and viral control, and no delay in viral rebound was observed after ART interruption, indicating that combination with other therapeutic strategies may be required [124].

#### 5.3.4. Therapies Harnessing Natural Killer Cells 

Most of the previously described immunotherapies assessed in NHP models and clinical trials targeted CD8+ or humoral cell responses [3]. In addition to CD8+ T cells, NK cells appear as a cell type of particular interest to harness for curative strategies. NK cells play an important role in the recognition and elimination of abnormal cells, including stressed or infected cells, and they have raised growing interest in the field of oncology [148,149,150,151] and infectious diseases [152,153,154]. NK cells clearly impact viremia levels [155]. NK cells are also play a role in controlling a viral reservoir in LNs. This has notably been brought to light in natural hosts, such as AGMs, in which NK cells migrate into LN follicles and mediate viral control [97]. In contrast, NK cells are generally not observed in follicles of SLTs in SIVmac-infected macaques [95,97]. SIVagm-infected AGMs exhibit high levels of CXCR5+ NK cells in SLT. Strategies that allow to increase the levels of NK cells in follicles during HIV infection could be useful and are currently under investigation in NHP models. It is also likely that in SIVagm infection, IL-15 enhances the survival of NK cells in follicles. Indeed, in AGM, the increase in NK cell numbers in follicles has been shown to be associated with a high production of IL-15 within follicles. IL-15 is presented there in membrane-bound form predominantly by FDC and eventually other antigen-presenting cells. IL-15, particularly in its membrane-bound form, is known to also enhance the cytotoxic profile of NK cells [156,157,158]. Accordingly, the use of IL-15 appears to be a potential way to target NK cell immunity for HIV clearance strategies. This cytokine was administered to SIVmac infected macaques and analyzed for its potential to promote NK cell activity and increase their access—as well as that of effector CD8+ T cells—to LN follicles. IL-15 stimulated NK cells have shown their ability to efficiently clear HIV-1-infected cells ex-vivo [158]. IL-15 and IL-15 superagonists that mimic the membrane-bound form, such as N-803, were initially considered for their latency reversing potential, which is still controversial. They might provide other benefits as immunotherapies, thanks to their immunomodulatory properties and their potential ability to increase the antiviral immune response and improve clearance of persistent infection. Several studies of the IL-15 superagonist N-803 have been conducted in NHP models. Webb et al. studied N-803 subcutaneous administration in SHIV-infected macaques and observed no evidence of latency reversal in SHIV-infected macaques, but they did reveal other interesting effects of this IL-15 superagonist [159]. Consistent with this work, another study on N-803 in SIV-infected ART-treated macaques also suggests that this IL-15 superagonist is not sufficient to exert an in vivo LRA effect when used alone [160,161]. The combination of N-803 with CD8β depletion was shown to be efficient in inducing virus reactivation in SHIV-infected ART-treated RMs [162]. Administration of the IL-15 superagonist may also be beneficial in activating and directing effector CD8+ T and NK cells to the B cell follicle [159]. This needs to be confirmed. IL-15 mediated activation of NK effector cells could also improve the ability of vaccine-induced antibody-dependent cellular cytotoxicity (ADCC) [163].

NK cell-mediated responses can be modulated in vivo by stimulation with IFN-α. Previous studies showed that IFN-α, when used in combination with ART, resulted in control of HIV in plasma and a decrease in integrated HIV DNA [164]. Similarly, in the study by Micci et al., a combination of recombinant IL-21 and pegylated-IFN-α2a significantly delayed viral rebound after ART cessation [165]. SIV-infected rhesus macaques receiving sequential interleukin-21 and IFN-α therapy generate terminally differentiated NK cells, which correlates with a reduction in replication-competent SIV in LN during ART and time to viral rebound after treatment interruption [166].

#### 5.3.5. Targeting Tissue Damage and Mucosal Immunity

Major challenges for HIV therapeutics arise from HIV-induced tissue damage, impaired mucosal immunity, immune exhaustion, and chronic inflammation. Tissue damage, such as collagen deposition in lymph nodes or loss of innate lymphoid cells and Th17 cells in intestinal mucosa, is induced by HIV infection already during primary infection. Tissue damage control and repair and/or disease tolerance are inherent features of the immune system [167], but are not sufficient to restore tissue and immune integrity and prevent excessive inflammation in HIV-infection. Therefore, therapeutic interventions aimed at favoring the protection of tissues and immune cells from the deleterious effects of HIV, repairing some of the induced damage, or reversing the chronic inflammatory state in treated HIV should participate at improving the quality of the anti-HIV immune responses [3,168]. For tissue repair and resolution of inflammation, several therapeutic approaches are being investigated. 

Interleukin-21 (IL-21) is a pleiotropic cytokine that exerts numerous enhancing and regulatory effects on immunity, which include maintaining CD8+ T cell functionality [169], promoting NK cell expansion [170], and inducing and preserving functional Th17 cell subsets [171]. In primates, pathogenic SIV infection of RMs is associated with a significant loss of IL-21-producing CD4+ T cells, which is not the case with nonpathogenic SIV infection of sooty mangabeys [83,104,172]. Because of its role in inducing and maintaining functional Th17, IL-21 was used to promote Th17 cells, with the goal that these Th17 cells would foster the intestinal barrier preservation and reduce inflammation. Pallikkuth et al. observed that SIV-infected RMs treated with IL-21 alone displayed transient increase in intestinal Th17 cells that was associated with reduced intestinal T cell proliferation, microbial translocation and decrease in systemic immune activation and inflammation. When combined with ART, Micci et al. demonstrated that IL-21 administration could lead to a reduction in residual inflammation and viral persistence in another SIV-infected macaque model [165]. Ortiz et al. subsequently showed that IL-21 in combination with probiotic therapy could improve Th17 frequencies while reducing markers of microbial translocation and dysbiosis, resulting in fewer comorbidities compared to controls in a similar NHP model [173]. More recently, the safety of a combination therapy consisting of IL-21 plus monoclonal antibody (mAb) against α4β7 has been assessed in uninfected RMs [174]. Combined IL-21 and anti-α4β7 mAb was well-tolerated and a reduction in gut homing of α4β7+ CD4 T cells was observed along with a decrease in the levels of gut immune activation [174]. 

The promising effects of IL-21 on Th17 cell preservation and reduced immune activation observed in NHP models are encouraging and suggest that immunotherapy with IL-21 may be beneficial for controlling tissue damage, restoring intestinal integrity, and enhancing immune responses [173]. 

Other agents, including modulators of innate immunity, inhibitors of proinflammatory cytokines, probiotic supplementation, anti-fibrotic and other molecules are also under investigation for their capacity to reduce systemic inflammation and/or tissue repair. 

### 5.4. Gene Editing and Gene Therapies

The field of gene editing is rapidly growing and characterized by tremendous progress in the recent past. Several strategies against other diseases are already approved in humans. In the field of HIV cure, the approaches aim for instance to excise the provirus out of the infected cell by using technologies such as CRISPR-Cas. Another strategy consists in removing the co-receptor CCR5. The expression of CCR5 is generally essential for allowing HIV to establish an infection. Promising results have been obtained by using zing-finger nucleases to remove CCR5 [175]. A combination of in vitro approaches, humanized mice models and NHP are used in the field to bring these approaches forward [176].

Chimeric antigen receptors (CAR) are hybrid antigen receptors with an extracellular antigen binding domain linked to an intracellular T cell activation domain [177]. CD4 and CD8 CAR T cells are constructed and evaluated for their capacity to reduce viral reservoir reduction in PLWH. Rust et al. studied the effects of virus-specific CD4-based CAR-T cells in SHIV-infected, ART-suppressed RMs [178]. The macaques were also infused with a boost of cell-associated HIV-1 envelope (Env). They observed a significant and unprecedented expansion of virus-specific CAR+ T-cells, as well as a significant delay in viral rebounds compared to controls after ART interruption. This is a remarkable proof-of-concept that CAR-T cells could be important players in combination therapies for HIV. Recently, humans stem cells (HSC)-derived CAR cells were shown to migrate to tissue-associated viral reservoirs and exhibit multilineage engraftment, persisting for 2 years in lymphoid germinal centers, brain, and intestine [179].

### 5.5. Approaches Targeting the Cells to the Right Place

HIV persistence in deep reservoirs and anatomical compartments that are inaccessible to the immune system or antiretroviral drugs is a major challenge for HIV therapeutic approaches. An increasing number of techniques and strategies are thus envisioned and tested to allow anti-HIV drugs and potent immune cells to cross these barriers to reach infected cells in tissues and sanctuary sites. 

As mentioned in the previous section, the IL-15 superagonist was shown to promote the migration of CD8+ T cells and NK cells into B cell follicles in SHIV-infected macaques [159]. On the other hand, using a NHP model, Ayala et al. demonstrated that transduction with CXCR5 by T cell engineering can lead to trafficking and localization of CD8 T cells into RM B-cell follicles [180]. 

A great variety of works are investigating new means and bioengineering techniques to deliver drugs to infected cells and/or present immunogens to immune cells in specific compartments, which could lead to enhanced efficiency of natural immune responses [181,182,183,184,185,186]. Pino et al. demonstrated in SIV-infected RM that administration of FTY720 (fingolimod), a drug used to treat multiple sclerosis, limits viral persistence by increasing the number of cytolytic cells in the LN, a critical site for HIV/SIV replication and persistence [187]. 

Martin et al. investigated two strategies to deliver HIV immunogens to B cell follicles in a RM model: first, in vivo formation of immune complexes (ICs) with a passively transferred anti-Env mAb, and second, generation of self-assembling protein nanoparticles displaying four copies of stabilized Env trimers [184]. They found that both ICs and nanoparticles led to a concentration of antigens at the periphery of B cell follicles in the draining LNs of NHPs. Nanoparticles were also shown to persist on FDC in the germinal centers. These encouraging results suggest that these two techniques could be used to target potent effector cells in B cell follicles. In a similar approach, Francica et al. designed synthetic “star” nanoparticles that target transport to LNs to deliver immunogen proteins on site [185]. They tested these nanoparticles with HIV-1 peptide minimal immunogens in both mice and NHP models. Although no neutralization was observed with the immunogens they used, their study represents an encouraging attempt for the use of nanoparticles in immunotherapies and vaccine strategies against HIV. Future work will certainly also expand to mRNA based delivery techniques.

## 6. Conclusions

The use of NHP models—to better understand lentiviral infection and pathogenesis and to evaluate curative approaches in in vivo preclinical models—has provided undisputable contributions to HIV science. The insights provided, for example on the viral reservoirs in distinct tissues during ART, are of major importance for HIV cure research. A great amount of promising results for HIV curative therapies have been obtained in NHP models. Of note, several of these results could not be reproduced in human clinical trials. However, this might not necessarily be due to species-specific differences, but rather due to differences in the protocols and schedules applied. Careful interpretation is needed, together with the continuous development of better tools. 

## Figures and Tables

**Figure 1 vaccines-09-00958-f001:**
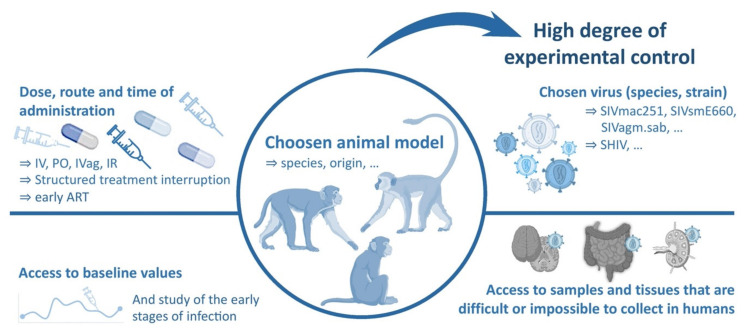
Interests and prerogatives of Non-Human Primates (NHP) models for HIV science. NHP models can be designed according to the question addressed, choosing primates from different species or origins that can be infected with different viruses. NHP models provide a means to study samples and tissues that are challenging to obtain and study in humans, as well as the possibility to access baseline values of the studied parameters and to study the earliest stages of infection after inoculation. Distinct times and routes of administration can be tested and analytical treatment interruptions (ATI) can be easily performed and monitored.

**Figure 2 vaccines-09-00958-f002:**
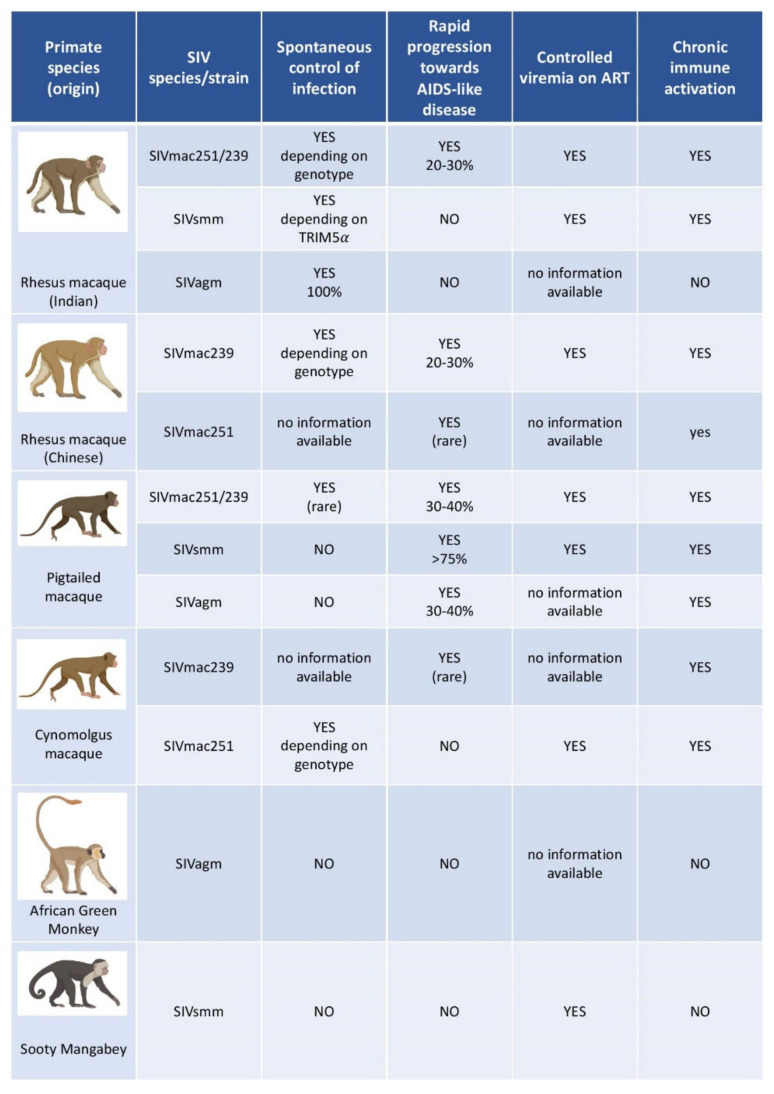
Main models of SIV infection in non-human primates studied in HIV research. This figure provides a non-exhaustive overview of non-human primates most often studied as models in HIV research. The principal features of each non-human species/viral strain are indicated. The viral load and other pathophysiological features vary depending on various parameters such as host species and genetic background, age, viral species or strain, laboratory, and sampling techniques, to mention just a few.

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
