# Peer review of "Interests of the Non-Human Primate Models for HIV Cure Research"

_vaccines, 2021, doi:10.3390/vaccines9090958_

Round 1
Reviewer 1 Report
Estimated Authors,
Estimated Editors,
I've read with great interest the present review from Terrade et al. on the Non-Human Primate model for HIV cure research.
The Authors have performed a considerable effort to summarize all of the available evidence on this topic, providing an up-to-date bibliography with appropriate references. Even the "casual reader" can follow quite easily the information flow across the text, and pros and cons for NPMs.
In summary: the completeness of the evidences provided by Terrade et al. represents a major plus for this review.
On the other hand, some minor issues should be addressed by the study Authors before its eventual acceptance.
1) as properly stated by Authors, NPM may be questioned for ethical and practical reasons. Therefore, why did the Authors include the more critical appraisal of NPM in the final section of the paper? The (self-)criticism and the reasons to overcome it may increase the interest and improve the understanding of the readers when placed in the initial section. Please note that this is a personal recommendation, not a formal requirement.
2) the section 2 to 5 deeply assess our understanding of both animal and viral models employed; even though this obviously improves the sum of information provided by the present article, it also risks to divert the reader from the main focus (i.e. pros and cons of the NPM). In other words, I would suggest to shorten and simplify the aforementioned section.
3) across the text, Authors often refer to the ART. This is obviously a main topic of the modern management of HIV/AIDS, but - at least in my opinion, Authors should limit their reference to ART when strictly necessary, as it does not represent the focus of this review (e.g. when discussing how animal model are somewhat impaired by the length of ART compared to the potential human hosts).
4) please be aware that the 3R are initially reported but explained only in the last section of the paper.
Reviewer 2 Report
The manuscript entitled “Interests of the Non-Human Primate Models for HIV Cure Research” by Terrade et al. is an extensive and interesting overview on the topic of non-human primate models for HIV cure research, and highlights the progress that has been made as well as limitations using these models. The review is very well written and well structured, and covers, to my knowledge, the necessary sub-categories. From this perspective, the article gives a timely update on this topic.
Therefore, after the revision of the following few minor points, I would suggest that the manuscript is suitable for publication in Vaccines.
Minor points:
Part 5, Page 6, line 220: the authors should please spell out the acronym PDC
Part 6, Page 7, line 272: the authors should please write “in vivo and in vitro” in italics
Part 6, Page 7, line 294: according to this reviewer the mentioned reference Barrenas et al. is not reported in the reference list. Please check.
Part 7.1.1, Page 8, line 359: the authors should please write the letter “a” of “CD8a” in italics
Part 7.3.5, Page 13, line 576: the authors should please spell out the acronym ILC
Part 9, Page 17, lines 805-806: the authors should please check the sentence “The use of NHP models – to better understand lentiviral infection and pathogenesis and have provided undisputable contributions to HIV science.”
